# Economic-Related Inequalities in Zero-Dose Children: A Study of Non-Receipt of Diphtheria–Tetanus–Pertussis Immunization Using Household Health Survey Data from 89 Low- and Middle-Income Countries

**DOI:** 10.3390/vaccines10040633

**Published:** 2022-04-18

**Authors:** Nicole Bergen, Bianca O. Cata-Preta, Anne Schlotheuber, Thiago M. Santos, M. Carolina Danovaro-Holliday, Tewodaj Mengistu, Samir V. Sodha, Daniel R. Hogan, Aluisio J. D. Barros, Ahmad Reza Hosseinpoor

**Affiliations:** 1Department of Data and Analytics, World Health Organization, 20 Avenue Appia, 1211 Geneva, Switzerland; bergenn@who.int (N.B.); schlotheuberan@who.int (A.S.); 2International Center for Equity in Health, Federal University of Pelotas, Rua Mal Deodoro 1160, Pelotas 96020-220, Brazil; bcatapreta@equidade.org (B.O.C.-P.); tmelo@equidade.org (T.M.S.); abarros@equidade.org (A.J.D.B.); 3Department of Immunization, Vaccines and Biologicals, World Health Organization, 20 Avenue Appia, 1211 Geneva, Switzerland; danovaroc@who.int (M.C.D.-H.); sodhas@who.int (S.V.S.); 4Gavi, The Vaccine Alliance, 40 Chemin du Pommier, 1218 Geneva, Switzerland; tmengistu@gavi.org (T.M.); dhogan@gavi.org (D.R.H.)

**Keywords:** inequality, socioeconomic factors, vaccination, immunization, global health, diphtheria–tetanus–pertussis (DTP) vaccine, Immunization Agenda 2030

## Abstract

Despite advances in scaling up new vaccines in low- and middle-income countries, the global number of unvaccinated children has remained high over the past decade. We used 2000–2019 household survey data from 154 surveys representing 89 low- and middle-income countries to assess within-country, economic-related inequality in the prevalence of one-year-old children with zero doses of diphtheria–tetanus–pertussis (DTP) vaccine. Zero-dose DTP prevalence data were disaggregated by household wealth quintile. Difference, ratio, slope index of inequality, concentration index, and excess change measures were calculated to assess the latest situation and change over time, by country income grouping for 17 countries with high zero-dose DTP numbers and prevalence. Across 89 countries, the median prevalence of zero-dose DTP was 7.6%. Within-country inequalities mostly favored the richest quintile, with 19 of 89 countries reporting a rich–poor gap of ≥20.0 percentage points. Low-income countries had higher inequality than lower–middle-income countries and upper–middle-income countries (difference between the median prevalence in the poorest and richest quintiles: 14.4, 8.9, and 2.7 percentage points, respectively). Zero-dose DTP prevalence among the poorest households of low-income countries declined between 2000 and 2009 and between 2010 and 2019, yet economic-related inequality remained high in many countries. Widespread economic-related inequalities in zero-dose DTP prevalence are particularly pronounced in low-income countries and have remained high over the previous decade.

## 1. Introduction

The Immunization Agenda 2030 (IA2030) envisions a world where everyone, everywhere, at every age, fully benefits from vaccines to improve health and well-being [1]. There have been notable gains in reaching children who have been partially vaccinated by making better use of existing vaccination delivery channels and addressing missed opportunities for vaccination among children who have some access to health services [2]. Yet, there are large numbers of children who have not received any routine vaccines (zero-dose children). The global number of zero-dose children has remained high over the past decade (following substantial reductions in the 1980s and again in the 2000s) [3] and increased following the onset of the COVID-19 pandemic [4,5]. Worldwide, the number of children who did not complete the three-dose diphtheria–tetanus–pertussis (DTP) series increased by 20% from 2019 to 2020 (zero-dose children accounted for 95% of the increased number), with three quarters of these children (17.1 million) not even receiving a first dose of DTP [6].

There are distinct measurement and programmatic considerations surrounding populations of zero-dose children versus those who are under-vaccinated (receiving some, but not all recommended vaccine doses on a national schedule). Whereas children who are under-vaccinated have had at least one contact with the health system, zero-dose children often lack access to basic health services, including immunization services. For instance, children who received fewer other health services and interventions, and whose mothers received fewer services (especially related to antenatal and delivery care), are at higher risk of being unvaccinated [7,8]. A study of immunization cascade pathways across low- and middle-income countries reported that three-quarters of one-year-old children who received a first dose of at least one of four basic vaccines universally recommended in infancy (Bacille Calmete-Guérin (BCG), DTP-containing, polio and measles-containing vaccines (MCV)) went on to receive all four [9].

Major global immunization initiatives have adopted an intensified focus on reaching zero-dose children. IA2030 includes an objective of extending immunization services to regularly reach zero-dose children (operationalized as children with zero doses of DTP), and makes a call for better data to map and track at the subnational level children who are unvaccinated [1]. Gavi 5.0 (the five year strategy of Gavi, the Vaccine Alliance, spanning 2021–2025) aims to reduce the number of zero-dose children by 25% by 2025, contributing to the IA2030 target of a 50% global reduction by 2030 [10]. The Equity Reference Group for Immunization (ERG), an advisory group of senior experts from UN organizations, not-for-profit organizations, academic institutions, and ministries of health, has strongly endorsed a focus on zero-dose children, noting that this is “an important and sensitive marker for communities that are at a disadvantage across a range of primary health care services and beyond” [11].

Few studies have characterized socioeconomic inequalities in zero-dose children. Across 92 low- and middle-income countries, an estimated 7.7% of one-year-old children did not receive any of four basic vaccines delivered in infancy, with higher zero-dose prevalence among the poorest (12.5%) than the richest (3.4%) wealth quintiles [9]. In their assessment of economic-related inequality in full vaccination coverage in 25 sub-Saharan African countries, Bobo et al. (2022) included a secondary outcome of unvaccinated status, reporting a concentration of zero-dose children in disadvantaged subgroups in most study countries [12]. In 2012, Bosch-Capblanch et al. assessed factors associated with non-vaccination across 96 low- and middle-income countries, reporting caregiver (and their partner) education levels, caregiver tetanus toxoid vaccination status, wealth index, and the type of family member involved in decision-making around illness to be the strongest predictors [13]; gender-related barriers also affect childhood immunization [14]. Johri et al. (2021) assessed the prevalence, distribution, and drivers of non-vaccination over a 24-year period in India, finding children with zero doses of DTP to be consistently more likely to belong to the poorest and least-educated families, and more likely to suffer from malnutrition [15]. In India, non-vaccination with DTP1 was also associated with low maternal education levels and fewer village healthcare resources [16].

To date, there have been no large multi-country studies that report the state of inequality for zero-dose children and that assess the latest status of inequality and change over time. In this study, we quantify the extent of economic-related inequality, including change in inequality over time, in zero-dose DTP prevalence, exploring variation by country income level grouping, and assessing patterns of inequality in countries with high prevalence or a high number of children with zero-dose DTP. Measuring and reporting socioeconomic inequalities in immunization can help inform strategies to equitably expand the reach of immunization programs and services.

## 2. Materials and Methods

### 2.1. Data

Disaggregated data were sourced from 88 Demographic Health Surveys (DHSs) and 66 Multiple Indicator Cluster Surveys (MICSs) in 89 low- and middle-income countries. DHS and MICS are large-scale, nationally representative household surveys that collect data through standardized face-to-face interviews with women aged 15–49 years, repeated approximately every three to five years. The data used in this analysis are the product of reanalysis of DHS and MICS microdata by the International Center for Equity in Health at the Federal University of Pelotas.

Zero-dose DTP prevalence was defined as the percentage of children aged 12–23 months who did not receive any of the three DTP-containing vaccine doses recommended in infancy at the time of the DHS or MICS. In certain countries, the sample specified different age ranges to align with national vaccination schedules and ensure that our indicator is consistent with results from national survey reports and recent publications on this topic. Surveys with an age range of 18–29 months are: Bosnia and Herzegovina, 2006 and 2011; Egypt, 2014; Jamaica, 2011; Peru, 2009 and 2019; Ukraine, 2012; Albania, 2008; Guyana, 2006; North Macedonia, 2005; Serbia, 2005; Suriname, 2006; and Tajikistan, 2005. Surveys with a 15–26-month age range are: Turkey, 2013; Kazakhstan, 2006; and Republic of Moldova, 2005 and 2012.

Household economic status was based on a household wealth index, which considered ownership of household assets and access to certain services. Principal component analysis was used to construct country-specific indices and generate wealth quintiles ranging from quintile 1 (poorest) to quintile 5 (richest). Information about wealth quintiles is available in the DHS or MICS datasets.

### 2.2. Country Selection

Countries were considered for inclusion based on the availability of data for children with zero-dose DTP from a DHS or MICS conducted between 2010 and 2019. The analysis included countries categorized by the World Bank Group as low-, lower–middle-, or upper–middle-income countries in July 2021. Countries included in the change over time analysis had additional data available from a survey conducted between 2000 and 2009 (allowing for a range of 5–15 years between the two surveys). Ten country surveys with an insufficient sample size (less than 25 for any subgroup) were excluded from the analysis.

A subset of 17 countries was identified for further analysis, based on (a) reporting more than 25% national zero-dose DTP prevalence according to the latest DHS or MICS between 2010 and 2019 or (b) being in the top 10 countries with the highest numbers of children with zero-dose DTP in 2019 [17].

### 2.3. Statistical Analysis

To assess the latest situation of inequality (using data from 2010–2019), we calculated summary measures of inequality using zero-dose DTP prevalence data disaggregated by wealth quintiles. Difference, a measure of absolute inequality, was calculated as prevalence in the poorest quintile minus prevalence in the richest quintile. Ratio, a measure of relative inequality, was calculated as prevalence in the poorest quintile divided by prevalence in the richest quintile.

Absolute and relative complex summary measures of inequality (slope index of inequality and concentration index, respectively), which take into account the situation across all wealth quintiles, were also calculated [18]. The slope index of inequality represents the difference in estimated zero-dose DTP prevalence between the poorest and richest quintiles using a regression model that accounts for the prevalence and weighting across all quintiles. The concentration index depicts the gradient across quintiles on a relative scale, indicating the extent to which zero-dose DTP is concentrated among the poorest subgroups. Concentration index accounts for the weighting of subgroups by population share. Negative values of the slope index of inequality and the concentration index indicate that zero-dose DTP is more prevalent among the poor than the rich.

To assess how inequalities have changed over time, annual absolute change was calculated using two data points, from 2010 to 2019 and from 2000 to 2009 (spanning about ten years). Annual absolute change was calculated as the difference in zero-dose DTP prevalence in the recent versus older surveys, divided by the number of years between the two surveys. It was calculated to quantify change in national average, and in the most and least advantaged subgroups.

Annual absolute excess change compares the annual absolute change in the least and most advantaged subgroups. For example, to arrive at absolute excess change for economic status, the annual absolute change in quintile 5 is subtracted from the annual absolute change in quintile 1, yielding annual absolute excess change in percentage points. A positive excess change indicates change favoring the richest subgroup (e.g., faster declines or slower increases in zero-dose DTP prevalence among quintile 5). A negative annual excess change indicates change favoring the poorest subgroup (e.g., faster declines or slower increases in zero-dose DTP prevalence among quintile 1).

A survey sampling design was taken into account when calculating point estimates of disaggregated data and corresponding 95% confidence intervals (95% CI). We ascertained statistical significance with 95% CIs.

Analyses were conducted using Stata 16 and Stata 17 software and data visuals were developed using Tableau software (version 2021.4).

## 3. Results

### 3.1. Latest Situation

The analysis of the latest situation of economic-related inequality included 89 countries, where the median national zero-dose DTP prevalence was 7.6%, ranging from 0% in Republic of Moldova to 72.7% in South Sudan (note: the South Sudan MICS 2010 occurred in the period leading up to South Sudan gaining independence in 2011) and 45.0% in Central African Republic (Table 1). While over a fifth of countries (19 of 89) showed a difference of 2.0 percentage points or less between zero-dose DTP prevalence in the poorest and richest quintiles, an equal number of countries (19 of 89) had a difference of at least 20 percentage points. Five countries showed higher zero-dose DTP prevalence among the richest than the poorest by a gap of over 5 percentage points, reaching a maximum difference of 10.7 and 10.1 percentage points in Tunisia and Namibia, respectively. The majority of countries (54 of 89) had prevalence that was at least twice as high in the poorest quintile than the richest.

Across 89 countries, the median zero-dose DTP prevalence among the poorest quintile (9.9%) was higher than the median prevalence among the richest quintile (5.0%) (Figure 1). Across all countries, the range of prevalence values in the richest quintile was smaller than the range of values in the poorer quintiles. In low-income countries (*n* = 22), the difference between the median prevalence in the poorest quintile (20.7%) and richest quintile (6.3%) was 14.4 percentage points. Lower–middle-income countries (*n* = 41) and upper–middle-income countries (*n* = 26) had comparatively smaller gaps, at 8.9 percentage points and 2.7 percentage points, respectively.

Slope index of inequality demonstrated similar patterns of inequality as difference across country income groupings, with the absolute extent of inequality tending to be slightly higher when taking all subgroups into account. Concentration index demonstrated similar patterns as ratio.

### 3.2. Change over Time

The change over time analysis included 65 countries with available data (Table 2). The median absolute annual change in national average across countries was −0.20 per year, indicating that, over a period of ten years, the national zero-dose DTP prevalence declined by 2.0 percentage points (median across 65 countries). Across countries, inequality hardly changed over time, with a low median change in both the poorest quintile (−0.16, or a decline of 1.6 percentage points over ten years) and the richest quintile (−0.04, or a decline of 0.4 percentage points over ten years).

Overall, changes in inequality and national average over time demonstrated different patterns across countries (Figure 2). Several countries showed the most desirable trend: decreased national zero-dose DTP prevalence, with larger improvements among the poorest households. Fourteen countries had a statistically significant decrease in absolute annual excess change. Most markedly, in Niger, the national zero-dose DTP prevalence decreased by 4.6 percentage points per year between 2006 and 2012, with much more reduction among the poorest quintile than the richest (absolute annual excess decline of 4.1 percentage points). Similarly, Burkina Faso, India, and Turkey decreased their national zero-dose DTP prevalence with narrowing inequality, while Namibia had no change in national prevalence alongside narrowing inequality (driven by increased zero-dose DTP prevalence in the richest quintile). Guinea and Madagascar both showed increased national zero-dose DTP prevalence driven by faster increases in the poorest quintile than the richest.

Many countries showed little or no absolute change in national average or economic-related inequality over the past ten years. Egypt and Mongolia sustained low zero-dose DTP prevalence (around 1% and 3%, respectively), with absolute economic-related inequality remaining around 2 percentage points or less. In other countries, such as Benin, high zero-dose DTP prevalence remained unchanged (above 15%), with a sustained gap of at least 25 percentage points between the poorest and richest quintiles.

In Congo, Côte d’Ivoire, and Haiti, the national zero-dose DTP prevalence remained largely unchanged between the two surveys (in the range of 14–20%), though the situation across wealth quintiles changed in divergent ways. In Haiti, the gap between the rich and poor widened due to increased prevalence among the poorest quintile (by more than 6 percentage points across ten years) and prevalence decreasing among the richest (by more than 7 percentage points across ten years). In Congo and Côte d’Ivoire, the zero-dose DTP prevalence increased among the richest quintile (by more than 9 percentage points across ten years) and decreased among the poorest quintile (by around 4–6 percentage points across ten years).

### 3.3. Countries with High Prevalence or Number of Children with Zero-Dose DTP

A subset of 17 countries had zero-dose DTP prevalence of more than 25% according to the latest available survey since 2010 (*n* = 13) and/or highest numbers of children with zero-dose DTP (*n* = 9). This subset consists of seven low-income countries, nine lower–middle-income countries, and one upper–middle-income country. All 17 countries reported higher prevalence in the poorest quintile than the richest, with difference ranging from 2.7 percentage points in Mexico to 53.2 percentage points in South Sudan (median difference of 32.1 percentage points across 17 countries).

Eleven of these countries had data from two or more surveys (Figure 3). Of these, four countries—Central African Republic, Guinea, Madagascar, and Philippines—had a statistically significant increase in national zero-dose DTP prevalence, with prevalence tending to increase in all wealth quintiles (except Central African Republic, where the prevalence among the richest quintile was unchanged). The gap between the richest and poorest increased by up to 12.2 percentage points over ten years in Guinea. Six countries had a statistically significant decrease in national prevalence. In India and Indonesia, there were large reductions in prevalence in the poorer subgroups which narrowed absolute inequality by 24.7 percentage points and 17.7 percentage points, respectively, over ten years. Nigeria and Pakistan reported improvements in all quintiles, with faster declines among the poorest than the richest, while Ethiopia reported faster improvements among the richest. In Lao People’s Democratic Republic, the zero-dose DTP prevalence declined among all except the poorest quintile.

## 4. Discussion

In this study, we described economic-related inequalities in zero-dose children across 89 low- and middle-income countries. Our results indicate that widespread economic-related inequalities to the detriment of the poorest countries, similar to previous multi-country analyses of inequalities in DTP3 coverage [19,20,21]. Our focus on zero-dose DTP prevalence, the flagship indicator of IA2030 and Gavi 5.0, yielded new insights into changes in inequality over time. However, a previous analysis of trends in economic-related inequality in DTP3 coverage showed narrowing inequality over the previous decade [20], while the present analysis of zero-dose DTP shows that economic-related inequalities have been largely sustained over the previous ten years. These divergent findings underscore the importance of dedicated monitoring and tracking the unvaccinated, using zero-dose DTP as the tracer indicator. The stark increase in the number of zero-dose children in 2020 further underscores the importance of the equity orientation of Gavi 5.0 and IA2030 and the need to restore, strengthen, and expand the reach of immunization systems.

Our study was the first, to our knowledge, to report socioeconomic inequality in zero-dose DTP across country income groupings. We reported more pronounced economic-related inequality in low-income countries than lower–middle- and upper–middle-income countries; however, there was variation across countries in the extent of inequality. The group of low-income countries demonstrated greater variation than lower–middle- or upper–middle-income countries. We found no apparent differences between country income groupings for changes in economic-related inequality over time. Previous research addressing trends in national DTP3 coverage in 46 African countries compared coverage in 2000 and 2015 by country income grouping and reported substantial improvements in the low-income countries; however, unlike our study, their analysis allowed for different country classification at the two time periods, and it did not report on socioeconomic inequality [22].

Multi-country studies such as ours are useful to describe global trends and inform where further studies and/or targeted resource distribution are needed. Our initial investigation of inequalities in a subset of 17 countries with a high prevalence or a high number of zero-dose children (including countries with low performance, but also better performing countries with large birth cohorts) revealed divergent patterns, particularly with regards to how inequality changed over time. Notably, India and Indonesia achieved substantial reductions in economic-related inequality, while the situation in Guinea and Madagascar became worse, in terms of both inequality and national average. Country-specific studies are required to explore contextual factors that underlie existing socioeconomic inequalities and identify lessons learned in areas where the situation is improving. While cross-cutting strategies to improve equity in vaccination coverage have been identified, including improving the relationships between the health system and mothers who are socioeconomically disadvantaged, setting-specific approaches should be tailored to account for local considerations [11,23].

Since the inception of the Expanded Program on Immunization in the 1970s, all countries have included DTP3 in the national vaccination schedules, as per the recommendation of the WHO. There are complex reasons why children from socioeconomically disadvantaged subgroups lack access to vaccinations. Women’s low social status is a barrier that cuts across family, community, and health facility environments, and manifests through educational attainment, income, decision-making, and resource allocation [14,24]. Deliberate measures are needed to mitigate women’s lack of access to immunization services and encourage greater involvement of fathers and communities in immunization initiatives. In sub-Saharan Africa, attributes related to parents or caregivers that are most frequently cited as barriers to vaccination include lack of knowledge about immunization, distance to access point, financial deprivation, lack of support from partner and distrust in vaccines, and immunization programs, whereas pertinent barriers related to health systems include human resource limitations, inadequate infrastructure, and supply chain inadequacies [25].

Others have explored the reasons underlying socioeconomic inequalities in vaccine coverage. For instance, across 46 low- and middle-income countries, pro-rich inequality in the receipt of four basic vaccines in childhood was associated with higher antenatal care visit attendance among wealthier women [26]. A study of country-level determinants reported national political stability, gender equality, and smaller land surface as factors that were predictive of more equitable DTP3 coverage in terms of wealth, education, and multidimensional poverty [27]. With a novel approach to measure multi-dimensional equity in immunization coverage, Patenaude and Odihi (2021) reported maternal education, socioeconomic status, and health insurance coverage as the primary drivers of unfair disadvantage in zero-dose status in India (accounting for 31%, 16%, and 7% of inequality, respectively) [28]. Further study of factors specific to “hard-to-reach” zero-dose children is warranted [29].

Deriving estimates and assessing inequalities of zero-dose children, a population defined by its lack of contact with health systems, is subject to measurement limitations [30]. Our use of household survey data helped to overcome inaccurate denominator values and absence of information about socioeconomic variables, i.e., major limitations of administrative data sources. Surveys, however, are subject to other limitations and biases, including exclusion of settings that are conflict-affected or hard to reach, respondent recall/self-reporting inaccuracies, and inaccurate or outdated sampling frames. Furthermore, children who were delayed in receiving vaccines may not have been captured. For these reasons, the reported estimates may underestimate zero-dose DTP prevalence.

Our analysis used a lack of DTP vaccination as a measure of zero-dose children, following the operational definition used by IA2030, with reports by World Health Organization and UNICEF [17] which can be measured with different data sources, and emphasized the reach of routine immunization services. Our study yielded similar results as a previous analysis that defined zero-dose children by the lack of vaccination of four basic vaccines universally recommended in infancy (BCG, DTP, MCV, and polio) [9].

## 5. Conclusions

Alongside indicators of vaccination coverage and drop-out, monitoring and reporting on zero-dose children is a stated priority of IA2030 and Gavi 5.0. The indicator provides valuable insights on children who lack access to routine immunizations and can help to inform actions to equitably expand the reach of immunization programs and basic healthcare services. In other words, actions should be equity-oriented to achieve national improvements alongside narrowing socioeconomic inequality (faster improvement among the poorest subgroups). To this end, our analysis of economic-related inequalities in zero-dose DTP prevalence serves as an important baseline assessment to benchmark future efforts to reduce the number of zero-dose children. Socioeconomic inequalities in zero-dose children, alongside other dimensions of inequality, should be monitored regularly as new data become available. Investments should expand and strengthen data collection and analysis, with a focus on disaggregated data. Further in-depth analysis of country-level inequalities is warranted, including exploration of the drivers of inequality and context-specific strategies to address them.

## Figures and Tables

**Figure 1 vaccines-10-00633-f001:**
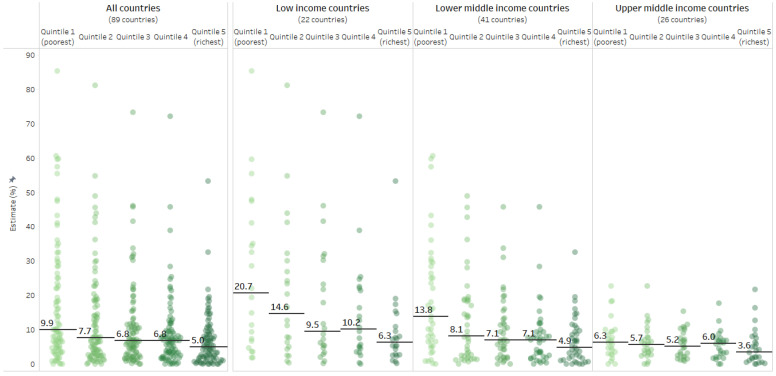
Percentage of children aged 12–23 months who did not receive any doses of DTP-containing vaccine in 89 countries, zero-dose DTP, disaggregated by economic status and grouped by country income level. Data taken from DHS and MICS 2010–2019. Circles indicate wealth quintile subgroups within countries; the horizontal lines indicate the median value across countries. Countries classified according to 2021 World Bank country income level groupings. DHS = Demographic and Health Survey. DTP = diphtheria–tetanus–pertussis vaccine. MICS = Multiple Indicator Cluster Survey.

**Figure 2 vaccines-10-00633-f002:**
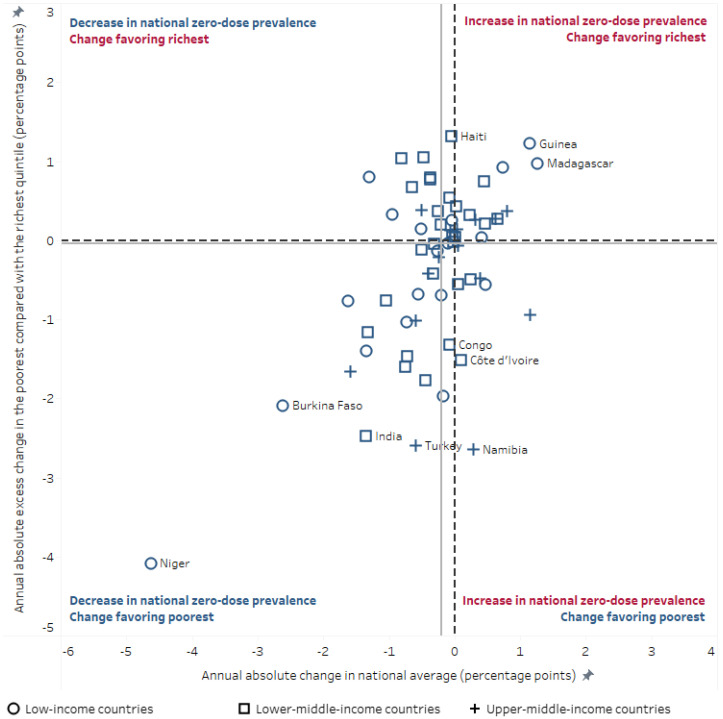
Change over time in national average percentage of children aged 12–23 months who did not receive any doses of DTP-containing vaccine, zero-dose DTP, and in wealth quintile 1 compared with wealth quintile 5, in 65 countries. Data taken from DHS and MICS 2000–2009 and 2010–2019. Every country is represented by a shape, which corresponds to its 2021 World Bank country income level grouping. For every study country, the annual absolute change in national average was calculated by subtracting the national average prevalence in survey year 1 (2000–2009) from prevalence in survey year 2 (2010–2019) and dividing by the number of intervening years. The annual absolute excess change was calculated by subtracting the annual absolute change in quintile 5 from the annual absolute change in quintile 1. The gray lines indicate the median values across countries. DHS = Demographic and Health Survey. DTP = diphtheria–tetanus–pertussis vaccine. MICS = Multiple Indicator Cluster Survey.

**Figure 3 vaccines-10-00633-f003:**
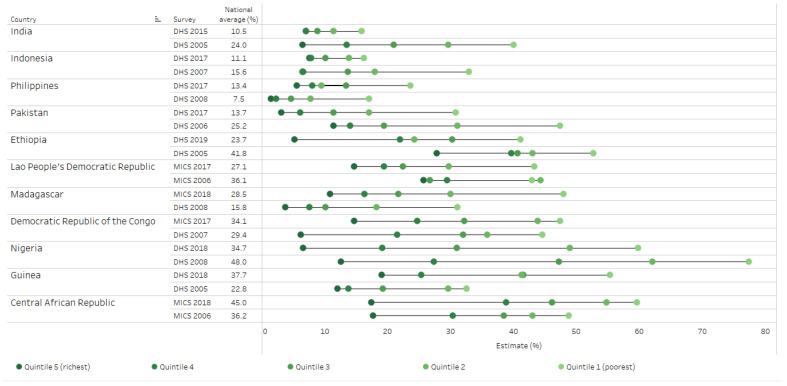
Time trends in percentage of children aged 12–23 months who did not receive any doses of DTP-containing vaccine in 11 countries, zero-dose DTP, by wealth quintile. Data taken from DHS and MICS 2000–2009 and 2010–2019. For each country, disaggregated data are presented for wealth quintiles by colored dots; the horizontal lines indicate the difference between the most extreme quintile values. DHS = Demographic and Health Survey. DTP = diphtheria–tetanus–pertussis vaccine. MICS = Multiple Indicator Cluster Survey.

**Table 1 vaccines-10-00633-t001:** Latest situation in percentage of children aged 12–23 months who did not receive any doses of DTP-containing vaccine in 89 low- and middle-income countries, zero-dose DTP, by wealth quintile (DHS and MICS 2010–2019).

Country	Source	National Average (95% CI)	Quintile 1 (Poorest) (95% CI)	Quintile 2 (95% CI)	Quintile 3 (95% CI)	Quintile 4 (95% CI)	Quintile 5 (Richest) (95% CI)	Difference (95% CI)	Ratio (95% CI)	Slope Index of Inequality (95% CI)	Concentration Index (95% CI)
Median (All Countries)		7.6 (4.3–10.9)	9.9 (5.0–14.9)	7.7 (3.7–11.8)	6.8 (3.4–10.2)	6.8 (4.0–9.7)	5.0 (2.9–7.1)	6.3 (2.6–10.0)	2.59 (0.19–4.98)	−7.8 (−12.2–−3.4)	−18.8 (−23.6–−14.1)
Low-Income Countries											
Median (Low-Income Countries)		11.9 (2.1–21.7)	20.7 (8.4–33.0)	14.6 (3.4–25.8)	9.5 (−1.1–20.2)	10.2 (1.3–19.1)	6.3 (0.2–12.4)	15.7 (8.0–23.4)	2.97 (−4.83–10.78)	−19.1 (−28.0–−10.2)	−19.5 (−25.5–−13.6)
Afghanistan	DHS 2015	27.0 (23.8–30.5)	34.3 (29.8–39.2)	32.3 (26.8–38.2)	31.3 (24.4–39.3)	22.6 (18.0–28.0)	15.3 (12.0–19.3)	19.1 (13.1–25.0)	2.25 (1.71–2.95)	−22.5 (−29.9–−15.0)	−14.6 (−19.8–−9.5)
Burkina Faso	DHS 2010	5.6 (4.5–7.1)	9.3 (6.5–13.3)	6.3 (4.4–9.1)	5.6 (3.9–7.8)	3.4 (2.0–5.8)	2.8 (1.3–5.7)	6.5 (2.6–10.5)	3.35 (1.48–7.56)	−7.8 (−12.2–−3.3)	−21.8 (−32.4–−11.1)
Burundi	DHS 2016	0.8 (0.5–1.2)	1.9 (0.9–3.6)	0.8 (0.3–2.0)	0.2 (0.0–1.3)	0.4 (0.1–1.5)	0.7 (0.3–2.0)	1.1 (−0.3–2.6)	2.59 (0.76–8.84)	−1.5 (−3.0–0.1)	−32.7 (−58.7–−6.7)
Central African Republic	MICS 2018	45.0 (41.7–48.3)	59.6 (52.9–66.0)	54.8 (48.0–61.5)	46.1 (39.6–52.8)	38.8 (33.1–44.9)	17.4 (13.3–22.5)	42.2 (34.2–50.2)	3.42 (2.57–4.56)	−44.9 (−53.8–−36.0)	−17.3 (−20.6–−13.9)
Democratic Republic of the Congo	MICS 2017	34.1 (30.2–38.2)	47.4 (41.1–53.8)	43.8 (37.4–50.4)	32.1 (25.6–39.4)	24.7 (17.6–33.5)	14.7 (10.3–20.6)	32.7 (24.5–40.9)	3.22 (2.22–4.68)	−38.4 (−48.5–−28.4)	−20.5 (−25.5–−15.5)
Ethiopia	DHS 2019	23.7 (19.2–28.9)	41.2 (30.4–52.8)	24.2 (16.3–34.3)	30.3 (20.5–42.3)	22.0 (13.3–34.1)	5.2 (2.2–12.0)	35.9 (23.8–48.1)	7.89 (3.22–19.33)	−39.5 (−52.6–−26.5)	−28.2 (−36.9–−19.5)
Gambia	MICS 2018	3.2 (2.2–4.7)	3.7 (1.9–6.9)	2.6 (1.1–6.0)	2.9 (1.4–5.9)	4.8 (1.9–11.4)	2.3 (0.6–8.5)	1.4 (−2.4–5.3)	1.63 (0.36–7.26)	−0.4 (−5.0–4.2)	1.3 (−20.6–23.2)
Guinea	DHS 2018	37.7 (34.1–41.4)	55.4 (47.2–63.2)	41.3 (35.1–47.7)	41.6 (34.6–48.8)	25.4 (19.9–31.8)	19.0 (13.6–26.0)	36.3 (26.2–46.5)	2.91 (2.04–4.15)	−41.5 (−52.2–−30.8)	−19.8 (−24.7–−15.0)
Guinea-Bissau	MICS 2018	7.0 (5.2–9.4)	7.4 (4.1–12.9)	7.8 (5.0–12.2)	3.3 (1.5–6.8)	9.6 (6.1–14.8)	7.6 (3.6–15.3)	−0.2 (−7.2–6.7)	0.97 (0.39–2.45)	0.6 (−7.2–8.5)	1.0 (−17.3–19.2)
Liberia	DHS 2019	8.6 (6.3–11.5)	11.4 (8.1–15.9)	7.7 (4.5–12.9)	8.7 (3.7–19.2)	10.7 (5.5–19.7)	0.2 (0.0–1.2)	11.3 (7.5–15.1)	70.75 (9.53–525.14)	−7.9 (−15.8–0.1)	−15.7 (−32.6–1.1)
Madagascar	MICS 2018	28.5 (25.9–31.2)	48.0 (43.2–52.8)	30.0 (25.5–34.8)	21.7 (17.5–26.5)	16.3 (12.1–21.6)	10.9 (7.6–15.3)	37.1 (31.0–43.3)	4.42 (3.08–6.34)	−44.8 (−51.9–−37.7)	−27.5 (−31.8–−23.3)
Malawi	DHS 2015	2.6 (1.9–3.5)	3.5 (2.0–5.9)	2.3 (1.3–4.1)	1.9 (1.0–3.5)	2.6 (1.3–5.1)	2.5 (1.2–5.2)	1.0 (−1.6–3.6)	1.39 (0.56–3.43)	−1.2 (−4.0–1.6)	−6.6 (−22.5–9.4)
Mali	DHS 2018	17.9 (15.1–21.1)	28.6 (21.7–36.7)	20.4 (15.2–26.7)	17.8 (13.6–22.9)	13.9 (10.0–19.1)	7.4 (4.8–11.2)	21.2 (13.1–29.4)	3.87 (2.36–6.35)	−23.7 (−33.2–−14.3)	−22.6 (−29.5–−15.7)
Mozambique	DHS 2015	10.0 (6.8–14.4)	19.4 (13.0–27.9)	12.7 (6.5–23.4)	5.4 (2.9–9.6)	4.5 (1.8–10.8)	2.7 (1.1–6.4)	16.7 (8.9–24.5)	7.09 (2.78–18.13)	−22.9 (−34.9–−11.0)	−35.7 (−46.0–−25.5)
Niger	DHS 2012	13.8 (11.6–16.5)	22.0 (17.4–27.4)	16.5 (12.3–21.8)	11.6 (8.4–16.0)	12.7 (8.1–19.4)	7.2 (4.9–10.7)	14.7 (9.0–20.5)	3.04 (1.93–4.78)	−15.8 (−22.7–−8.9)	−20.6 (−28.0–−13.2)
Rwanda	DHS 2014	0.9 (0.5–1.7)	1.9 (0.8–4.5)	0.4 (0.1–2.9)	0.9 (0.2–3.9)	0.0 (0.0–0.0)	1.1 (0.3–3.2)	0.9 (−1.2–2.9)	1.81 (0.44–7.42)	−1.3 (−3.5–1.0)	−25.5 (−61.7–10.8)
Sierra Leone	DHS 2019	5.4 (4.2–6.9)	6.6 (4.4–9.9)	4.4 (2.5–7.5)	7.2 (4.6–11.0)	3.5 (1.8–6.7)	4.4 (2.0–9.5)	2.3 (−2.1–6.6)	1.52 (0.62–3.68)	−2.6 (−7.2–2.1)	−6.7 (−20.5–7.1)
South Sudan *	MICS 2010	72.7 (69.6–75.5)	85.3 (79.7–89.6)	81.2 (76.2–85.4)	73.3 (67.1–78.8)	72.2 (66.5–77.3)	53.2 (46.4–59.9)	32.1 (23.8–40.5)	1.60 (1.40–1.84)	−36.2 (−45.5–−26.9)	−8.6 (−10.9–−6.3)
Sudan	MICS 2014	16.8 (14.3–19.6)	35.0 (28.1–42.5)	23.4 (18.8–28.7)	10.4 (7.6–13.9)	7.5 (5.0–11.0)	5.4 (3.0–9.5)	29.6 (21.8–37.5)	6.53 (3.51–12.15)	−38.2 (−47.5–−28.9)	−36.7 (−43.7–−29.7)
Togo	MICS 2017	9.2 (6.8–12.2)	14.9 (10.4–20.9)	11.1 (5.9–19.9)	6.2 (3.2–11.8)	5.2 (2.1–12.1)	8.0 (4.0–15.3)	6.9 (−0.6–14.4)	1.86 (0.87–3.96)	−10.4 (−19.8–−1.0)	−16.5 (−34.0–0.9)
Uganda	DHS 2016	5.1 (4.1–6.3)	4.7 (3.2–6.9)	5.1 (3.3–7.7)	4.7 (2.8–7.8)	5.6 (3.6–8.7)	5.4 (3.2–9.0)	−0.6 (−4.0–2.7)	0.88 (0.46–1.68)	0.9 (−2.9–4.6)	1.7 (−10.7–14.1)
Yemen	DHS 2013	23.4 (21.2–25.8)	32.5 (26.9–38.6)	26.8 (22.8–31.3)	23.3 (19.1–28.0)	21.4 (17.2–26.2)	9.5 (7.0–12.9)	23.0 (16.4–29.5)	3.41 (2.39–4.87)	−24.6 (−32.4–−16.8)	−19.2 (−24.2–−14.3)
**Lower–Middle-Income Countries**											
**Median** **(Lower–Middle-Income Countries)**		**9.4** **(5.3–13.4)**	**13.8** **(7.2–20.4)**	**8.1** **(2.8–13.4)**	**7.1** **(3.0–11.1)**	**7.1** **(3.8–10.4)**	**4.9** **(2.1–7.6)**	**8.4** **(2.7–14.1)**	**3.23** **(1.26–5.20)**	**−11.3** **(−18.2–−4.4)**	**−23.8** **(−31.7–−15.9)**
Algeria	MICS 2018	4.5 (3.7–5.5)	6.9 (5.0–9.5)	3.1 (2.0–4.6)	5.2 (3.3–8.3)	3.4 (2.0–5.9)	3.3 (1.6–6.4)	3.6 (0.5–6.8)	2.12 (1.00–4.49)	−3.8 (−7.6–0.0)	−15.7 (−28.5–−3.0)
Angola	DHS 2015	31.2 (28.6–34.0)	57.5 (51.6–63.2)	42.7 (37.9–47.6)	21.7 (17.6–26.4)	12.1 (7.6–18.6)	9.4 (5.8–14.8)	48.1 (40.8–55.4)	6.13 (3.81–9.86)	−59.7 (−67.1–−52.4)	−33.4 (−37.9–−28.9)
Belize	MICS 2015	7.1 (4.4–11.2)	10.2 (4.7–21.0)	1.5 (0.5–4.5)	1.9 (0.5–7.7)	7.1 (1.8–23.8)	16.1 (7.2–32.3)	−5.9 (−20.3–8.6)	0.64 (0.22–1.86)	6.0 (−10.6–22.6)	10.7 (−23.1–44.5)
Benin	DHS 2017	15.8 (13.8–17.9)	32.4 (26.8–38.6)	18.2 (14.7–22.3)	11.8 (9.1–15.3)	8.8 (6.3–12.0)	6.9 (4.6–10.2)	25.5 (19.1–32.0)	4.71 (3.05–7.27)	−30.3 (−37.7–−22.9)	−32.6 (−38.8–−26.4)
Cambodia	DHS 2014	6.0 (4.6–7.9)	11.7 (8.3–16.2)	8.1 (5.0–13.0)	6.9 (3.5–13.2)	0.8 (0.2–2.9)	0.6 (0.2–1.7)	11.0 (7.1–15.0)	18.82 (6.37–55.59)	−17.1 (−23.3–−10.9)	−43.2 (−53.4–−33.1)
Cameroon	DHS 2018	16.7 (14.2–19.5)	29.9 (23.8–36.8)	18.6 (13.7–24.7)	15.6 (11.9–20.2)	10.0 (6.8–14.6)	4.9 (2.8–8.2)	25.0 (18.0–32.0)	6.15 (3.45–10.97)	−28.1 (−36.3–−19.9)	−29.1 (−35.7–−22.6)
Comoros	DHS 2012	17.8 (14.1–22.2)	26.4 (16.9–38.7)	22.6 (15.2–32.4)	11.6 (6.8–18.9)	11.8 (5.8–22.3)	14.5 (7.9–25.1)	11.9 (−1.9–25.7)	1.82 (0.89–3.72)	−17.6 (−31.3–−3.9)	−15.7 (−28.4–−2.9)
Congo	MICS 2014	14.0 (12.0–16.3)	28.5 (23.9–33.4)	14.1 (10.6–18.6)	7.2 (4.2–12.1)	7.4 (4.2–12.8)	8.7 (4.7–15.7)	19.7 (12.6–26.8)	3.26 (1.74–6.11)	−25.4 (−33.5–−17.4)	−32.6 (−42.1–−23.0)
Côte d’Ivoire	MICS 2016	19.6 (17.3–22.2)	29.6 (24.7–34.9)	18.9 (14.7–23.9)	19.7 (15.2–25.1)	11.8 (7.6–17.9)	11.3 (6.4–19.0)	18.3 (10.3–26.3)	2.63 (1.48–4.66)	−22.2 (−30.9–−13.5)	−19.2 (−26.8–−11.7)
Egypt	DHS 2014	0.6 (0.3–1.1)	1.2 (0.4–4.0)	0.7 (0.2–2.4)	0.5 (0.2–1.6)	0.3 (0.1–1.2)	0.3 (0.0–1.7)	0.9 (−0.6–2.5)	4.23 (0.49–36.52)	−1.0 (−2.5–0.5)	−33.6 (−68.2–1.0)
El Salvador	MICS 2014	1.1 (0.5–2.5)	0.2 (0.0–1.5)	0.9 (0.3–2.9)	1.1 (0.4–2.7)	3.3 (0.8–12.1)	0.0 (0.0–0.0)	0.2 (−0.2–0.6)	NA	1.5 (−1.3–4.2)	19.4 (−11.5–50.3)
Eswatini	MICS 2014	3.4 (2.1–5.6)	4.7 (1.9–11.3)	1.3 (0.3–5.3)	1.9 (0.5–7.4)	7.6 (3.0–18.0)	2.6 (0.7–8.8)	2.1 (−3.3–7.4)	1.78 (0.39–8.23)	0.4 (−6.2–6.9)	−3.2 (−37.6–31.2)
Ghana	MICS 2017	4.0 (2.8–5.8)	5.4 (2.9–9.8)	5.2 (2.5–10.4)	4.1 (2.2–7.8)	3.3 (1.4–7.3)	2.2 (0.9–5.5)	3.2 (−0.7–7.0)	2.45 (0.81–7.44)	−4.4 (−9.7–1.0)	−17.1 (−34.4–0.2)
Haiti	DHS 2016	16.5 (13.3–20.1)	25.1 (18.9–32.5)	19.6 (13.1–28.3)	16.1 (11.3–22.4)	9.4 (5.4–15.7)	4.4 (1.3–14.1)	20.7 (12.1–29.3)	5.73 (1.64–20.05)	−25.5 (−36.3–−14.6)	−25.5 (−34.7–−16.3)
Honduras	DHS 2011	0.9 (0.5–1.5)	0.8 (0.2–2.6)	1.3 (0.6–3.2)	1.2 (0.4–3.3)	0.9 (0.3–3.0)	0.0 (0.0–0.0)	0.8 (−0.2–1.8)	NA	−0.7 (−2.1–0.7)	−13.4 (−39.7–12.8)
India	DHS 2015	10.5 (10.1–10.9)	15.8 (15.0–16.7)	11.4 (10.5–12.2)	8.8 (8.0–9.7)	7.0 (6.2–7.9)	7.1 (5.8–8.5)	8.8 (7.2–10.4)	2.24 (1.84–2.74)	−11.5 (−13.2–−9.8)	−18.9 (−21.7–−16.1)
Indonesia	DHS 2017	11.1 (9.8–12.7)	16.2 (13.3–19.7)	13.9 (10.9–17.5)	10.2 (7.7–13.3)	7.9 (5.6–10.9)	7.6 (5.4–10.6)	8.6 (4.6–12.7)	2.14 (1.45–3.15)	−12.0 (−17.0–−6.9)	−18.6 (−25.6–−11.6)
Kenya	DHS 2014	2.5 (1.9–3.3)	5.2 (3.8–6.9)	1.6 (0.9–2.8)	1.7 (0.5–5.8)	0.8 (0.4–1.6)	2.2 (0.8–5.9)	3.0 (0.3–5.7)	2.38 (0.83–6.86)	−3.8 (−6.8–−0.8)	−27.9 (−51.9–−3.9)
Kiribati	MICS 2018	40.1 (35.1–45.4)	40.4 (29.3–52.5)	36.3 (26.7–47.0)	45.8 (35.2–56.7)	45.8 (35.1–56.9)	32.6 (22.0–45.4)	7.7 (−8.9–24.4)	1.24 (0.78–1.97)	−3.0 (−21.7–15.6)	−1.3 (−9.0–6.4)
Kyrgyzstan	MICS 2018	9.4 (6.7–12.9)	10.2 (4.9–20.0)	4.2 (1.8–9.6)	7.1 (3.1–15.1)	9.3 (4.7–17.5)	18.3 (11.1–28.7)	−8.1 (−19.4–3.2)	0.56 (0.24–1.30)	8.7 (−3.0–20.4)	18.0 (−0.7–36.8)
Lao People’s Democratic Republic	MICS 2017	27.1 (24.7–29.7)	43.3 (38.0–48.7)	29.7 (24.9–35.1)	22.4 (18.1–27.4)	19.5 (15.2–24.5)	14.6 (10.8–19.5)	28.7 (21.7–35.6)	2.96 (2.14–4.08)	−34.5 (−42.3–−26.6)	−22.1 (−26.9–−17.2)
Lesotho	MICS 2018	8.5 (6.3–11.2)	17.0 (11.0–25.3)	7.4 (3.6–14.8)	5.5 (2.5–11.5)	3.7 (1.7–8.0)	8.5 (3.8–18.2)	8.4 (−1.3–18.2)	1.99 (0.81–4.86)	−11.5 (−22.0–−1.1)	−19.8 (−37.7–−1.8)
Mauritania	MICS 2015	14.2 (12.1–16.6)	25.0 (19.0–32.2)	18.5 (14.6–23.3)	13.0 (9.6–17.4)	8.1 (5.3–12.1)	4.1 (1.8–8.9)	20.9 (13.6–28.3)	6.10 (2.66–14.01)	−27.7 (−36.8–−18.6)	−31.3 (−39.8–−22.8)
Mongolia	MICS 2018	3.0 (1.9–4.5)	3.6 (1.8–6.9)	2.0 (0.9–4.3)	4.1 (1.8–9.3)	3.2 (1.2–7.8)	1.6 (0.4–6.3)	2.0 (−1.3–5.3)	2.25 (0.48–10.55)	−1.1 (−4.9–2.6)	−3.2 (−25.8–19.4)
Myanmar	DHS 2015	13.1 (10.1–16.9)	18.1 (12.1–26.1)	18.9 (12.7–27.2)	18.3 (10.7–29.4)	2.2 (0.7–7.0)	3.7 (1.2–10.9)	14.4 (6.3–22.5)	4.88 (1.51–15.74)	−21.6 (−32.5–−10.6)	−27.0 (−38.8–−15.3)
Nepal	MICS 2019	10.5 (8.5–13.0)	13.8 (10.1–18.7)	14.5 (10.1–20.4)	9.4 (5.5–15.4)	7.1 (3.9–12.4)	6.6 (3.6–12.0)	7.2 (1.3–13.1)	2.09 (1.06–4.13)	−11.1 (−18.2–−4.1)	−16.8 (−27.2–−6.4)
Nigeria	DHS 2018	34.7 (32.7–36.7)	59.8 (56.0–63.4)	48.9 (44.8–53.1)	31.0 (27.7–34.5)	19.1 (16.3–22.3)	6.6 (4.9–8.8)	53.2 (49.0–57.4)	9.08 (6.69–12.32)	−62.4 (−66.9–−57.9)	−32.5 (−35.0–−30.0)
Pakistan	DHS 2017	13.7 (11.2–16.5)	30.8 (23.7–38.9)	17.1 (12.2–23.3)	11.4 (7.9–16.3)	6.1 (3.8–9.7)	3.1 (1.6–6.0)	27.6 (19.7–35.6)	9.85 (4.85–20.03)	−36.1 (−46.5–−25.7)	−39.7 (−47.1–−32.2)
Papua New Guinea	DHS 2016	36.1 (32.2–40.2)	60.7 (51.9–68.9)	45.6 (37.7–53.9)	33.8 (27.6–40.6)	28.4 (22.7–34.9)	13.4 (9.1–19.1)	47.3 (37.4–57.2)	4.54 (3.06–6.75)	−51.9 (−61.7–−42.1)	−25.7 (−30.3–−21.1)
Philippines	DHS 2017	13.4 (11.5–15.5)	23.6 (19.8–28.0)	9.4 (6.5–13.5)	13.4 (8.8–19.7)	8.0 (4.6–13.4)	5.6 (2.6–11.5)	18.0 (12.2–23.9)	4.22 (1.97–9.07)	−20.1 (−27.3–−12.9)	−26.4 (−35.2–−17.5)
Republic of Moldova	MICS 2012	0.0 (0.0–0.0)	0.0 (0.0–0.0) **	0.0 (0.0–0.0)	0.0 (0.0–0.0)	0.0 (0.0–0.0)	0.0 (0.0–0.0)	0.0 (0.0–0.0)	NA		
Sao Tome and Principe	MICS 2019	2.3 (1.1–4.7)	6.5 (2.7–14.8)	1.3 (0.2–8.8)	0.0 (0.0–0.0)	2.0 (0.3–12.4)	0.0 (0.0–0.0)	6.5 (1.0–12.0)	NA	−8.3 (−19.7–3.1)	−47.0 (−91.0–−3.0)
Senegal	DHS 2019	3.8 (2.6–5.6)	8.9 (5.3–14.6)	2.3 (1.0–5.5)	4.5 (1.6–12.0)	1.9 (0.6–5.5)	0.5 (0.1–2.4)	8.4 (3.8–13.0)	17.91 (3.44–93.24)	−9.3 (−15.8–−2.9)	−41.9 (−61.2–−22.7)
Tajikistan	DHS 2017	7.6 (6.1–9.4)	8.1 (5.1–12.6)	6.5 (3.8–10.9)	6.4 (4.2–9.8)	7.7 (4.9–11.7)	10.5 (7.3–14.8)	−2.4 (−7.6–2.8)	0.77 (0.43–1.38)	2.3 (−3.2–7.9)	5.1 (−7.0–17.1)
Timor-Leste	DHS 2016	21.6 (18.8–24.7)	36.0 (29.2–43.4)	28.1 (22.4–34.6)	19.8 (14.9–25.9)	15.4 (11.5–20.4)	9.8 (5.8–15.9)	26.2 (17.6–34.9)	3.69 (2.15–6.32)	−30.7 (−39.8–−21.6)	−25.6 (−32.7–−18.6)
Tunisia	MICS 2018	4.6 (3.1–6.7)	0.8 (0.1–5.4)	1.1 (0.3–4.4)	6.7 (3.3–13.2)	4.2 (1.8–9.5)	11.5 (6.4–19.7)	−10.7 (−17.3–−4.1)	0.07 (0.01–0.52)	12.4 (4.9–19.9)	45.4 (30.4–60.4)
Ukraine	MICS 2012	15.6 (12.1–20.0)	22.1 (14.5–32.2)	13.5 (6.8–25.1)	10.4 (5.8–18.2)	13.0 (6.3–25.0)	19.5 (11.8–30.7)	2.6 (−10.3–15.5)	1.13 (0.61–2.12)	−1.5 (−15.8–12.9)	−0.0 (−15.6–15.5)
United Republic of Tanzania	DHS 2015	3.0 (2.2–4.2)	6.6 (4.4–9.7)	3.6 (1.9–6.7)	2.7 (1.4–5.1)	0.3 (0.1–1.4)	1.2 (0.4–3.8)	5.4 (2.5–8.3)	5.55 (1.62–18.96)	−7.8 (−11.8–−3.9)	−41.4 (−55.5–−27.2)
Viet Nam	MICS 2013	3.8 (2.6–5.5)	12.7 (8.2–19.3)	2.6 (1.0–6.9)	2.0 (0.5–8.0)	1.7 (0.4–7.1)	0.7 (0.1–5.0)	12.0 (6.4–17.7)	17.98 (2.39–135.16)	−12.8 (−19.8–−5.8)	−55.3 (−77.2–−33.5)
Zambia	DHS 2018	2.1 (1.4–3.1)	3.1 (1.8–5.5)	2.5 (1.2–5.1)	1.8 (0.8–3.7)	1.6 (0.5–5.1)	1.0 (0.2–4.1)	2.2 (−0.1–4.4)	3.23 (0.67–15.59)	−2.7 (−5.6–0.2)	−22.2 (−43.0–−1.3)
Zimbabwe	MICS 2019	5.5 (3.9–7.6)	8.0 (4.3–14.4)	7.8 (4.3–13.7)	5.8 (3.0–10.8)	2.5 (1.0–6.0)	1.7 (0.6–4.6)	6.3 (1.2–11.4)	4.65 (1.47–14.75)	−9.1 (−16.1–−2.0)	−26.4 (−40.9–−11.9)
**Upper–Middle-Income Countries**											
**Median** **(Upper–Middle-Income Countries)**		**4.8** **(2.7–6.9)**	**6.3** **(3.5–9.1)**	**5.7** **(3.2–8.1)**	**5.2** **(3.2–7.1)**	**6.0** **(4.1–8.0)**	**3.6** **(0.9–6.3)**	**2.4** **(−0.0–4.8)**	**1.92** **(−1.12–4.95)**	**−3.6** **(−6.9–−0.4)**	**−9.1** **(−17.7–−0.5)**
Albania	DHS 2017	0.5 (0.1–1.8)	0.9 (0.1–6.2)	0.0 (0.0–0.0)	1.1 (0.2–7.5)	0.0 (0.0–0.0)	0.0 (0.0–0.0) **	0.9 (−0.9–2.7)	NA	−20.8 (−40.5–−1.0)	−5.2 (−9.7–−0.8)
Armenia	DHS 2015	1.5 (0.7–3.3)	1.6 (0.4–6.2)	2.4 (0.3–14.9)	2.2 (0.7–7.0)	2.0 (0.5–8.0)	0.0 (0.0–0.0)	1.6 (−0.6–3.7)	NA	−2.1 (−5.4–1.2)	−26.3 (−55.9–3.4)
Bosnia and Herzegovina	MICS 2011	3.0 (1.8–4.9)	4.7 (1.7–12.3)	5.7 (2.5–12.4)	0.8 (0.1–5.7)	3.0 (1.1–8.2)	1.6 (0.4–6.1)	3.1 (−2.1–8.2)	2.99 (0.54–16.57)	−4.3 (−10.3–1.6)	−25.5 (−57.7–6.7)
Costa Rica	MICS 2018	2.3 (1.1–4.7)	4.1 (2.2–7.6)	3.2 (0.8–12.1)	1.2 (0.3–5.3)	0.1 (0.0–0.6)	2.0 (0.5–7.4)	2.1 (−1.6–5.8)	2.03 (0.47–8.64)	−5.2 (−9.3–−1.1)	−33.6 (−51.1–−16.0)
Cuba	MICS 2019	2.6 (1.3–5.0)	4.7 (1.5–14.0)	4.3 (1.2–14.0)	1.6 (0.4–6.0)	1.3 (0.3–4.7)	0.3 (0.0–2.0)	4.4 (−0.9–9.8)	17.28 (1.77–169.02)	−6.4 (−13.8–1.1)	−36.2 (−57.2–−15.1)
Dominican Republic	MICS 2014	8.9 (7.7–10.2)	14.1 (11.5–17.1)	9.6 (7.0–13.0)	5.5 (3.8–7.9)	6.9 (4.8–9.8)	6.3 (4.1–9.7)	7.7 (3.8–11.7)	2.22 (1.38–3.60)	−10.0 (−14.6–−5.4)	−20.0 (−27.7–−12.3)
Gabon	DHS 2012	11.6 (9.0–14.7)	18.1 (13.9–23.3)	12.2 (7.5–19.2)	9.1 (4.8–16.4)	7.6 (3.1–17.5)	8.5 (2.3–26.5)	9.7 (−1.8–21.1)	2.14 (0.60–7.56)	−12.9 (−24.0–−1.8)	−18.8 (−34.6–−3.0)
Guatemala	DHS 2014	2.5 (1.8–3.3)	2.8 (1.7–4.6)	3.0 (1.7–5.3)	2.2 (1.2–3.8)	2.0 (0.7–5.5)	2.0 (0.8–5.3)	0.8 (−1.6–3.2)	1.38 (0.47–4.08)	−1.3 (−3.9–1.3)	−10.4 (−27.0–6.2)
Guyana	MICS 2014	4.4 (2.7–7.2)	6.3 (3.6–10.7)	2.7 (1.0–7.2)	6.5 (2.0–19.5)	3.2 (1.1–9.0)	1.9 (0.5–7.3)	4.4 (0.2–8.7)	3.40 (0.76–15.17)	−3.9 (−9.4–1.6)	−10.8 (−31.7–10.1)
Iraq	MICS 2018	13.3 (11.4–15.5)	18.4 (15.0–22.4)	14.0 (10.5–18.4)	15.4 (11.0–21.2)	12.5 (8.2–18.5)	5.0 (2.9–8.4)	13.4 (8.8–18.0)	3.70 (2.08–6.56)	−13.2 (−19.0–−7.4)	−18.5 (−25.9–−11.2)
Jamaica	MICS 2011	4.8 (2.5–9.3)	5.7 (1.6–17.9)	0.6 (0.1–4.3)	10.0 (2.9–28.8)	6.5 (2.3–16.7)	0.9 (0.1–6.6)	4.7 (−2.4–11.8)	6.08 (0.59–62.25)	−0.8 (−8.3–6.6)	−6.2 (−33.5–21.2)
Jordan	DHS 2017	7.4 (5.7–9.7)	9.4 (6.3–13.9)	7.9 (4.8–12.8)	4.7 (2.6–8.4)	7.5 (3.4–15.8)	7.5 (2.5–20.0)	2.0 (−6.7–10.6)	1.26 (0.41–3.86)	−3.8 (−12.2–4.5)	−7.8 (−25.4–9.7)
Kazakhstan	MICS 2015	4.4 (3.2–6.0)	3.6 (1.6–7.6)	2.4 (1.1–5.0)	2.9 (1.4–5.9)	6.3 (3.5–10.9)	8.0 (3.9–15.7)	−4.4 (−10.6–1.8)	0.45 (0.16–1.27)	5.8 (−0.3–12.0)	19.2 (−0.0–38.4)
Maldives	DHS 2016	9.2 (6.6–12.8)	9.5 (6.0–14.8)	7.8 (4.5–13.1)	10.9 (6.8–17.0)	1.7 (0.5–5.7)	16.4 (6.7–34.8) **	−6.9 (−21.2–7.4)	0.58 (0.23–1.50)	2.2 (−10.2–14.5)	4.3 (−18.2–26.9)
Mexico	MICS 2015	7.8 (5.6–10.7)	8.5 (4.2–16.8)	6.3 (3.8–10.1)	10.6 (5.9–18.2)	6.9 (3.5–13.2)	5.8 (1.7–18.0)	2.7 (−6.4–11.9)	1.47 (0.37–5.84)	−1.3 (−10.5–7.9)	−1.6 (−19.9–16.8)
Namibia	DHS 2013	7.3 (5.5–9.7)	2.6 (1.2–5.6)	7.0 (3.5–13.7)	6.8 (3.6–12.4)	8.8 (4.2–17.7)	12.7 (7.4–21.0)	−10.1 (−17.1–−3.1)	0.21 (0.08–0.53)	10.2 (2.5–17.9)	24.0 (9.3–38.8)
North Macedonia	MICS 2018	4.1 (1.7–9.5)	0.0 (0.0–0.0)	12.9 (4.3–32.7)	3.1 (0.7–12.3)	5.9 (1.1–25.6)	0.6 (0.1–4.3)	−0.6 (−1.7–0.6)	0.00	−3.5 (−12.2–5.3)	−11.7 (−34.0–10.5)
Panama	MICS 2013	7.7 (5.5–10.8)	10.1 (6.3–15.6)	5.6 (2.4–12.9)	10.4 (5.2–19.7)	6.2 (1.9–18.1)	2.0 (0.6–6.8)	8.0 (2.9–13.2)	5.00 (1.34–18.70)	−5.4 (−15.0–4.1)	−7.7 (−27.2–11.9)
Paraguay	MICS 2016	5.2 (3.9–6.9)	6.8 (4.2–11.1)	3.7 (1.6–8.6)	2.6 (1.1–5.8)	8.2 (5.1–13.2)	3.8 (1.6–8.7)	3.1 (−1.6–7.7)	1.81 (0.68–4.82)	−0.4 (−5.8–5.0)	−3.4 (−20.6–13.8)
Peru	DHS 2019	4.7 (3.9–5.7)	6.3 (4.7–8.3)	4.0 (2.6–6.0)	4.9 (3.1–7.6)	3.3 (2.1–5.4)	5.0 (2.6–9.2)	1.3 (−2.3–4.9)	1.26 (0.63–2.51)	−2.0 (−5.4–1.3)	−6.6 (−18.8–5.6)
Serbia	MICS 2019	3.5 (1.8–6.7)	7.6 (2.9–18.6)	6.5 (1.7–21.9)	1.5 (0.2–10.6)	4.5 (1.1–16.9)	0.5 (0.1–3.9)	7.1 (−0.1–14.3)	14.01 (1.59–123.87)	−8.4 (−17.0–0.2)	−41.4 (−68.6–−14.3)
South Africa	DHS 2016	8.8 (6.4–12.2)	9.9 (5.4–17.5)	10.7 (6.1–18.1)	6.3 (3.0–13.0)	6.8 (2.8–16.0)	10.1 (3.3–26.8)	−0.1 (−12.3–12.0)	0.99 (0.29–3.30)	−3.0 (−14.2–8.2)	−5.6 (−27.7–16.5)
Suriname	MICS 2018	19.7 (15.5–24.7)	22.7 (16.3–30.8)	22.7 (15.7–31.6)	11.6 (6.2–20.5)	17.7 (9.0–31.7)	21.6 (10.8–38.8)	1.1 (−14.7–16.9)	1.05 (0.51–2.16)	−6.9 (−22.7–8.9)	−4.8 (−18.4–8.8)
Thailand	MICS 2019	3.1 (1.8–5.4)	3.1 (1.8–5.4)	4.1 (1.1–14.4)	3.6 (1.0–12.2)	3.8 (1.9–7.5)	0.1 (0.1–0.4)	2.9 (1.2–4.7)	22.11 (6.88–71.01)	−2.4 (−6.2–1.4)	−11.1 (−25.2–3.0)
Tonga	MICS 2019	3.6 (1.4–8.8)	0.0 (0.0–0.0)	0.0 (0.0–0.0) **	7.1 (1.5–27.2) **	9.7 (2.2–34.5) **	4.2 (1.0–16.2) **	−4.2 (−10.0–1.7)	0.00	10.8 (0.1–21.4)	39.8 (16.5–63.1)
Turkey	DHS 2013	5.6 (3.7–8.3)	7.2 (3.9–12.8)	6.5 (2.9–13.9)	7.5 (2.9–17.8)	1.7 (0.5–5.2)	3.3 (0.9–11.5)	3.9 (−2.1–9.8)	2.16 (0.53–8.75)	−5.5 (−12.1–1.0)	−20.9 (−41.2–−0.7)

Note: Difference calculated by subtracting the prevalence in quintile 5 (richest) from the prevalence in quintile 1 (poorest). Ratio calculated by dividing the prevalence in quintile 1 by the prevalence in quintile 5. Countries classified according to 2021 World Bank country income level groupings. DHS = Demographic and Health Survey. DTP = diphtheria–tetanus–pertussis vaccine. CI = confidence interval. MICS = Multiple Indicator Cluster Survey. NA = not applicable. * Survey occurred in the period leading up to South Sudan gaining independence in 2011. ** Estimate is based on 25–49 cases.

**Table 2 vaccines-10-00633-t002:** Change over time in percentage of children aged 12–23 months who did not receive any doses of DTP-containing vaccine in 65 low- and middle-income countries, zero-dose DTP, by wealth quintile (DHS and MICS 2000–2019).

			National Average			Quintile 1 (Poorest)			Quintile 5 (Richest)			Excess Change
Country	Source 1	Source 2	Year 1 (95% CI)	Year 2 (95% CI)	Absolute Annual Change (95% CI)	Year 1 (95% CI)	Year 2 (95% CI)	Absolute Annual Change (95% CI)	Year 1 (95% CI)	Year 2 (95% CI)	Absolute Annual Change (95% CI)	Absolute Annual Excess Change (95% CI)
Median (All Countries)			10⋅2 (6.6–13.8)	7.3 (4.1–10.5)	−0.20 (−0.47–0.07)	13.1 (7.3–18.8)	9.4 (4.2–14.6)	−0.16 (−0.53–0.22)	4.3 (2.2–6.4)	4.4 (2.8–6.0)	−0.04 (−0.26–0.18)	−0.03 (−0.38–0.31)
Low-Income Countries												
Median (Low-Income Countries)			15.8 (8.8–22.8)	9.2 (0.9–17.4)	−0.26 (−1.05–0.54)	26.1 (16.7–35.5)	14.9 (3.1–26.7)	−0.44 (−1.40–0.51)	6.7 (2.6–10.8)	5.4 (2.2–8.6)	−0.04 (−0.49–0.41)	−0.13 (−0.88–0.62)
Burkina Faso	DHS 2003	DHS 2010	23.9 (19.9–28.5)	5.6 (4.5–7.1)	−2.61 (−3.25–−1.97)	32.9 (25.7–41.1)	9.3 (6.5–13.3)	−3.37 (−4.58–−2.17)	11.8 (7.6–17.9)	2.8 (1.3–5.7)	−1.28 (−2.06–−0.51)	−2.09 (−3.52–−0.65)
Burundi	MICS 2005	DHS 2016	11.2 (9.2–13.5)	0.8 (0.5–1.2)	−0.94 (−1.14–−0.75)	10.8 (7.4–15.6)	1.9 (0.9–3.6)	−0.82 (−1.20–−0.43)	13.3 (8.5–20.3)	0.7 (0.3–2.0)	−1.14 (−1.68–−0.61)	0.33 (−0.33–0.98)
Central African Republic	MICS 2006	MICS 2018	36.2 (31.8–40.8)	45.0 (41.7–48.3)	0.73 (0.27–1.20)	48.8 (39.6–58.0)	59.6 (52.9–66.0)	0.90 (−0.05–1.85)	17.7 (12.0–25.3)	17.4 (13.3–22.5)	−0.02 (−0.69–0.65)	0.92 (−0.24–2.09)
Democratic Republic of the Congo	DHS 2007	MICS 2017	29.4 (25.1–34.1)	34.1 (30.2–38.2)	0.46 (−0.13–1.06)	44.6 (36.2–53.3)	47.4 (41.1–53.8)	0.28 (−0.79–1.35)	6.2 (3.7–10.3)	14.7 (10.3–20.6)	0.85 (0.24–1.45)	−0.57 (−1.80–0.66)
Ethiopia	DHS 2005	DHS 2019	41.8 (37.8–45.9)	23.7 (19.2–28.9)	−1.29 (−1.74–−0.84)	52.6 (46.3–58.9)	41.2 (30.4–52.8)	−0.82 (−1.75–0.11)	27.8 (20.4–36.8)	5.2 (2.2–12.0)	−1.62 (−2.28–−0.95)	0.79 (−0.35–1.94)
Gambia	MICS 2005	MICS 2018	3.7 (2.8–5.0)	3.2 (2.2–4.7)	−0.04 (−0.17–0.09)	2.4 (1.3–4.6)	3.7 (1.9–6.9)	0.10 (−0.12–0.31)	4.3 (2.2–8.0)	2.3 (0.6–8.5)	−0.15 (−0.47–0.16)	0.25 (−0.13–0.63)
Guinea	DHS 2005	DHS 2018	22.8 (19.3–26.6)	37.7 (34.1–41.4)	1.15 (0.75–1.54)	32.5 (23.8–42.7)	55.4 (47.2–63.2)	1.76 (0.80–2.71)	12.1 (7.0–20.0)	19.0 (13.6–26.0)	0.54 (−0.14–1.22)	1.22 (0.05–2.39)
Guinea-Bissau	MICS 2006	MICS 2018	15.8 (13.2–18.8)	7.0 (5.2–9.4)	−0.73 (−1.02–−0.44)	19.0 (13.4–26.1)	7.4 (4.1–12.9)	−0.96 (−1.59–−0.33)	6.7 (3.4–12.8)	7.6 (3.6–15.3)	0.07 (−0.52–0.67)	−1.04 (−1.90–−0.17)
Liberia	DHS 2007	DHS 2019	24.7 (20.4–29.5)	8.6 (6.3–11.5)	−1.34 (−1.77–−0.91)	39.3 (30.8–48.5)	11.4 (8.1–15.9)	−2.32 (−3.12–−1.52)	11.2 (6.2–19.3)	0.2 (0.0–1.2)	−0.92 (−1.45–−0.39)	−1.41 (−2.37–−0.44)
Madagascar	DHS 2008	MICS 2018	15.8 (13.7–18.2)	28.5 (25.9–31.2)	1.27 (0.92–1.62)	31.1 (26.1–36.6)	48.0 (43.2–52.8)	1.69 (0.98–2.40)	3.7 (1.9–7.1)	10.9 (7.6–15.3)	0.71 (0.26–1.16)	0.98 (0.13–1.82)
Malawi	MICS 2006	DHS 2015	3.6 (2.9–4.4)	2.6 (1.9–3.5)	−0.11 (−0.23–0.02)	4.1 (2.6–6.5)	3.5 (2.0–5.9)	−0.07 (−0.37–0.22)	2.9 (1.7–4.8)	2.5 (1.2–5.2)	−0.04 (−0.30–0.23)	−0.03 (−0.43–0.36)
Mali	MICS 2009	DHS 2018	14.2 (12.7–15.9)	17.9 (15.1–21.1)	0.41 (0.04–0.78)	25.9 (21.9–30.3)	28.6 (21.7–36.7)	0.31 (−0.65–1.27)	5.0 (3.0–8.1)	7.4 (4.8–11.2)	0.27 (−0.17–0.71)	0.04 (−1.02–1.09)
Mozambique	DHS 2003	DHS 2015	12.4 (10.1–15.2)	10.0 (6.8–14.4)	−0.20 (−0.58–0.17)	26.1 (20.2–32.9)	19.4 (13.0–27.9)	−0.56 (−1.37–0.26)	1.0 (0.2–4.9)	2.7 (1.1–6.4)	0.14 (−0.09–0.38)	−0.70 (−1.55–0.15)
Niger	DHS 2006	DHS 2012	41.6 (37.6–45.7)	13.8 (11.6–16.5)	−4.63 (−5.42–−3.84)	54.1 (44.9–63.0)	22.0 (17.4–27.4)	−5.35 (−7.07–−3.63)	14.8 (10.8–20.0)	7.2 (4.9–10.7)	−1.27 (−2.16–−0.37)	−4.08 (−6.02–−2.14)
Rwanda	DHS 2005	DHS 2014	3.2 (2.0–5.2)	0.9 (0.5–1.7)	−0.26 (−0.44–−0.08)	4.0 (1.6–9.2)	1.9 (0.8–4.5)	−0.23 (−0.65–0.19)	1.9 (0.9–4.1)	1.1 (0.3–3.2)	−0.09 (−0.30–0.12)	−0.13 (−0.60–0.34)
Sierra Leone	DHS 2008	DHS 2019	23.2 (19.9–26.8)	5.4 (4.2–6.9)	−1.62 (−1.96–−1.29)	25.7 (18.7–34.1)	6.6 (4.4–9.9)	−1.73 (−2.47–−0.98)	14.9 (9.7–22.2)	4.4 (2.0–9.5)	−0.96 (−1.60–−0.31)	−0.77 (−1.76–0.21)
Togo	MICS 2006	MICS 2017	15.3 (12.3–18.9)	9.2 (6.8–12.2)	−0.56 (−0.94–−0.18)	19.0 (13.0–26.8)	14.9 (10.4–20.9)	−0.37 (−1.15–0.41)	4.6 (1.9–10.8)	8.0 (4.0–15.3)	0.31 (−0.30–0.92)	−0.68 (−1.68–0.31)
Uganda	DHS 2006	DHS 2016	10.2 (8.5–12.2)	5.1 (4.1–6.3)	−0.51 (−0.72–−0.30)	9.1 (6.1–13.4)	4.7 (3.2–6.9)	−0.44 (−0.84–−0.04)	11.2 (7.5–16.3)	5.4 (3.2–9.0)	−0.58 (−1.09–−0.07)	0.14 (−0.51–0.79)
Yemen	MICS 2006	DHS 2013	24.6 (20.0–29.9)	23.4 (21.2–25.8)	−0.17 (−0.95–0.60)	40.4 (30.8–50.8)	32.5 (26.9–38.6)	−1.14 (−2.80–0.52)	3.7 (1.5–8.7)	9.5 (7.0–12.9)	0.83 (0.21–1.46)	−1.97 (−3.74–−0.19)
**Lower–Middle-Income Countries**												
**Median** **(Lower–Middle-Income Countries)**			**9.0** **(3.9–14.1)**	**8.0** **(4.2–11.9)**	**−0.23** **(−0.45–−0.01)**	**16.8** **(8.8–24.9)**	**13.3** **(6.6–20.0)**	**−0.10** **(−0.49–0.29)**	**4.1** **(1.4–6.8)**	**3.6** **(1.8–5.4)**	**−0.13** **(−0.38–0.12)**	**0.05** **(−0.39–0.49)**
Benin	DHS 2006	DHS 2017	16.0 (14.3–17.9)	15.8 (13.8–17.9)	−0.02 (−0.27–0.22)	30.2 (26.0–34.7)	32.4 (26.8–38.6)	0.20 (−0.46–0.87)	4.4 (2.8–6.9)	6.9 (4.6–10.2)	0.23 (−0.08–0.53)	−0.02 (−0.76–0.71)
Cambodia	DHS 2005	DHS 2014	9.4 (7.7–11.6)	6.0 (4.6–7.9)	−0.38 (−0.66–−0.10)	13.1 (9.4–17.8)	11.7 (8.3–16.2)	−0.16 (−0.79–0.48)	9.0 (5.0–15.7)	0.6 (0.2–1.7)	−0.93 (−1.51–−0.35)	0.77 (−0.08–1.63)
Cameroon	MICS 2006	DHS 2018	11.3 (9.0–14.2)	16.7 (14.2–19.5)	0.45 (0.14–0.75)	19.5 (13.7–27.0)	29.9 (23.8–36.8)	0.87 (0.09–1.64)	3.4 (1.2–9.3)	4.9 (2.8–8.2)	0.12 (−0.24–0.49)	0.74 (−0.11–1.60)
Congo	DHS 2005	MICS 2014	14.7 (11.3–18.9)	14.0 (12.0–16.3)	−0.08 (−0.56–0.40)	31.9 (22.3–43.3)	28.5 (23.9–33.4)	−0.38 (−1.66–0.89)	0.3 (0.0–2.1)	8.7 (4.7–15.7)	0.94 (0.35–1.53)	−1.32 (−2.73–0.08)
Côte d’Ivoire	MICS 2006	MICS 2016	18.7 (14.9–23.0)	19.6 (17.3–22.2)	0.10 (−0.37–0.57)	35.3 (26.7–45.0)	29.6 (24.7–34.9)	−0.58 (−1.63–0.48)	1.9 (0.7–5.3)	11.3 (6.4–19.0)	0.94 (0.29–1.58)	−1.51 (−2.75–−0.27)
Egypt	DHS 2005	DHS 2014	0.9 (0.3–2.4)	0.6 (0.3–1.1)	−0.03 (−0.09–0.03)	1.0 (0.1–8.0)	1.2 (0.4–4.0)	0.03 (−0.17–0.22)	0.6 (0.0–11.4)	0.3 (0.0–1.7)	−0.04 (−0.15–0.07)	0.06 (−0.16–0.29)
Eswatini	DHS 2006	MICS 2014	4.0 (2.4–6.7)	3.4 (2.1–5.6)	−0.08 (−0.41–0.25)	3.6 (1.3–9.2)	4.7 (1.9–11.3)	0.14 (−0.54–0.82)	5.8 (2.0–15.6)	2.6 (0.7–8.8)	−0.40 (−1.25–0.44)	0.54 (−0.54–1.62)
Ghana	DHS 2008	MICS 2017	2.0 (1.1–3.6)	4.0 (2.8–5.8)	0.23 (0.02–0.44)	1.6 (0.5–5.6)	5.4 (2.9–9.8)	0.42 (−0.01–0.84)	1.3 (0.2–9.3)	2.2 (0.9–5.5)	0.09 (−0.28–0.47)	0.32 (−0.24–0.89)
Haiti	DHS 2005	DHS 2016	17.0 (13.9–20.6)	16.5 (13.3–20.1)	−0.05 (−0.48–0.39)	18.5 (14.0–24.0)	25.1 (18.9–32.5)	0.60 (−0.17–1.36)	12.2 (6.8–21.0)	4.4 (1.3–14.1)	−0.71 (−1.51–0.08)	1.31 (0.21–2.41)
Honduras	DHS 2005	DHS 2011	0.8 (0.4–1.5)	0.9 (0.5–1.5)	0.02 (−0.10–0.13)	0.4 (0.1–1.5)	0.8 (0.2–2.6)	0.07 (−0.11–0.25)	2.2 (0.7–6.8)	0.0 (0.0–0.0)	−0.36 (−0.78–0.06)	0.43 (−0.03–0.89)
India	DHS 2005	DHS 2015	24.0 (22.6–25.5)	10.5 (10.1–10.9)	−1.35 (−1.51–−1.20)	40.0 (36.7–43.3)	15.8 (15.0–16.7)	−2.42 (−2.76–−2.07)	6.5 (5.1–8.2)	7.1 (5.8–8.5)	0.06 (−0.15–0.27)	−2.47 (−2.87–−2.07)
Indonesia	DHS 2007	DHS 2017	15.6 (13.9–17.5)	11.1 (9.8–12.7)	−0.45 (−0.68–−0.21)	32.9 (28.7–37.4)	16.2 (13.3–19.7)	−1.67 (−2.21–−1.13)	6.6 (4.4–9.7)	7.6 (5.4–10.6)	0.10 (−0.26–0.46)	−1.77 (−2.42–−1.12)
Kenya	DHS 2003	DHS 2014	10.8 (8.4–13.7)	2.5 (1.9–3.3)	−0.75 (−1.00–−0.51)	25.7 (18.6–34.3)	5.2 (3.8–6.9)	−1.86 (−2.59–−1.14)	5.1 (2.6–9.8)	2.2 (0.8–5.9)	−0.27 (−0.63–0.10)	−1.60 (−2.41–−0.78)
Lao People’s Democratic Republic	MICS 2006	MICS 2017	36.1 (30.6–41.9)	27.1 (24.7–29.7)	−0.81 (−1.37–−0.25)	42.9 (33.8–52.6)	43.3 (38.0–48.7)	0.03 (−0.96–1.02)	25.7 (15.2–40.0)	14.6 (10.8–19.5)	−1.01 (−2.20–0.19)	1.04 (−0.52–2.59)
Lesotho	DHS 2009	MICS 2018	4.3 (3.0–6.1)	8.5 (6.3–11.2)	0.46 (0.15–0.78)	9.0 (5.7–13.9)	17.0 (11.0–25.3)	0.89 (−0.01–1.79)	2.4 (0.6–9.2)	8.5 (3.8–18.2)	0.68 (−0.16–1.51)	0.21 (−1.02–1.44)
Mauritania	MICS 2007	MICS 2015	19.3 (16.8–22.1)	14.2 (12.1–16.6)	−0.64 (−1.08–−0.21)	29.3 (23.7–35.6)	25.0 (19.0–32.2)	−0.53 (−1.64–0.59)	13.7 (9.5–19.5)	4.1 (1.8–8.9)	−1.20 (−1.94–−0.46)	0.68 (−0.66–2.01)
Mongolia	MICS 2005	MICS 2018	2.9 (1.7–4.9)	3.0 (1.9–4.5)	0.00 (−0.15–0.16)	4.5 (2.0–10.1)	3.6 (1.8–6.9)	−0.07 (−0.41–0.27)	3.0 (0.7–12.2)	1.6 (0.4–6.3)	−0.11 (−0.49–0.27)	0.04 (−0.47–0.55)
Nepal	DHS 2006	MICS 2019	7.3 (5.1–10.5)	10.5 (8.5–13.0)	0.25 (−0.02–0.51)	16.6 (10.5–25.1)	13.8 (10.1–18.7)	−0.21 (−0.85–0.43)	3.0 (0.7–12.4)	6.6 (3.6–12.0)	0.28 (−0.17–0.74)	−0.49 (−1.28–0.29)
Nigeria	DHS 2008	DHS 2018	48.0 (45.7–50.2)	34.7 (32.7–36.7)	−1.33 (−1.63–−1.03)	77.4 (74.0–80.4)	59.8 (56.0–63.4)	−1.76 (−2.25–−1.27)	12.5 (10.1–15.5)	6.6 (4.9–8.8)	−0.60 (−0.93–−0.27)	−1.17 (−1.75–−0.58)
Pakistan	DHS 2006	DHS 2017	25.2 (22.7–27.9)	13.7 (11.2–16.5)	−1.05 (−1.39–−0.71)	47.4 (40.8–54.2)	30.8 (23.7–38.9)	−1.51 (−2.44–−0.59)	11.4 (7.6–16.8)	3.1 (1.6–6.0)	−0.75 (−1.21–−0.30)	−0.76 (−1.79–0.27)
Philippines	DHS 2008	DHS 2017	7.5 (6.1–9.3)	13.4 (11.5–15.5)	0.65 (0.37–0.94)	17.1 (13.0–22.1)	23.6 (19.8–28.0)	0.73 (0.05–1.40)	1.5 (0.3–6.7)	5.6 (2.6–11.5)	0.46 (−0.07–0.98)	0.27 (−0.59–1.13)
Republic of Moldova	DHS 2005	MICS 2012	1.7 (0.7–4.1)	0.0 (0.0–0.0)	−0.25 (−0.46–−0.03)	0.0 (0.0–0.0) *	0.0 (0.0–0.0) *	0.00 (0.00–0.00)	2.6 (0.6–10.2)	0.0 (0.0–0.0)	−0.37 (−0.88–0.14)	0.37 (−0.14–0.88)
Sao Tome and Principe	DHS 2008	MICS 2019	6.4 (4.0–10.0)	2.3 (1.1–4.7)	−0.37 (−0.67–−0.07)	6.0 (2.6–13.4)	6.5 (2.7–14.8)	0.05 (−0.62–0.72)	8.1 (3.0–20.2)	0.0 (0.0–0.0)	−0.74 (−1.44–−0.04)	0.79 (−0.18–1.76)
Senegal	DHS 2005	DHS 2019	6.8 (5.4–8.6)	3.8 (2.6–5.6)	−0.21 (−0.37–−0.06)	8.2 (5.5–11.9)	8.9 (5.3–14.6)	0.05 (−0.34–0.45)	2.5 (0.9–6.5)	0.5 (0.1–2.4)	−0.14 (−0.32–0.04)	0.19 (−0.24–0.63)
Tajikistan	MICS 2005	DHS 2017	7.0 (4.9–9.9)	7.6 (6.1–9.4)	0.05 (−0.20–0.30)	9.6 (4.9–17.9)	8.1 (5.1–12.6)	−0.12 (−0.72–0.48)	5.3 (2.6–10.5)	10.5 (7.3–14.8)	0.43 (−0.00–0.87)	−0.55 (−1.30–0.19)
Timor-Leste	DHS 2009	DHS 2016	24.9 (22.2–27.9)	21.6 (18.8–24.7)	−0.47 (−1.05–0.11)	35.6 (30.2–41.4)	36.0 (29.2–43.4)	0.05 (−1.24–1.35)	16.7 (12.2–22.5)	9.8 (5.8–15.9)	−0.99 (−2.00–0.02)	1.05 (−0.60–2.69)
United Republic of Tanzania	DHS 2004	DHS 2015	6.7 (4.9–9.0)	3.0 (2.2–4.2)	−0.33 (−0.53–−0.12)	12.4 (8.2–18.3)	6.6 (4.4–9.7)	−0.53 (−1.04–−0.02)	2.4 (0.6–8.7)	1.2 (0.4–3.8)	−0.11 (−0.42–0.20)	−0.42 (−1.01–0.18)
Viet Nam	DHS 2002	MICS 2013	11.7 (8.3–16.3)	3.8 (2.6–5.5)	−0.72 (−1.10–−0.34)	28.1 (19.4–38.9)	12.7 (8.2–19.3)	−1.40 (−2.41–−0.39)	0.0 (0.0–0.0)	0.7 (0.1–5.0)	0.06 (−0.06–0.19)	−1.46 (−2.48–−0.44)
Zambia	DHS 2007	DHS 2018	7.7 (5.9–10.0)	2.1 (1.4–3.1)	−0.51 (−0.71–−0.31)	7.2 (4.5–11.5)	3.1 (1.8–5.5)	−0.37 (−0.72–−0.02)	3.7 (1.6–8.4)	1.0 (0.2–4.1)	−0.25 (−0.56–0.06)	−0.12 (−0.58–0.34)
Zimbabwe	MICS 2009	MICS 2019	8.6 (7.0–10.6)	5.5 (3.9–7.6)	−0.31 (−0.57–−0.05)	12.3 (8.7–17.3)	8.0 (4.3–14.4)	−0.43 (−1.08–0.21)	5.6 (2.8–10.8)	1.7 (0.6–4.6)	−0.38 (−0.80–0.03)	−0.05 (−0.81–0.72)
**Upper–Middle-Income Countries**												
**Median** **(Upper–Middle-Income Countries)**			**5.2** **(0.6–9.7)**	**4.6** **(1.6–7.6)**	**0.02** **(−0.38–0.42)**	**8.5** **(0.3–16.6)**	**6.3** **(2.3–10.2)**	**−0.05** **(−0.78–0.68)**	**2.3** **(−2.1–6.6)**	**4.2** **(0.1–8.3)**	**0.06** **(−0.48–0.61)**	**−0.14** **(−0.80–0.52)**
Albania	DHS 2008	DHS 2017	0.5 (0.1–3.4)	0.5 (0.1–1.8)	0.00 (−0.12–0.12)	0.0 (0.0–0.0)	0.9 (0.1–6.2)	0.10 (−0.10–0.30)	0.0 (0.0–0.0) *	0.0 (0.0–0.0) *	0.00 (0.00–0.00)	0.10 (−0.10–0.30)
Armenia	DHS 2005	DHS 2015	5.5 (2.9–10.0)	1.5 (0.7–3.3)	−0.39 (−0.74–−0.04)	6.5 (2.1–18.4)	1.6 (0.4–6.2)	−0.49 (−1.23–0.24)	0.7 (0.1–5.1) *	0.0 (0.0–0.0)	−0.07 (−0.21–0.07)	−0.42 (−1.17–0.32)
Bosnia and Herzegovina	MICS 2006	MICS 2011	6.0 (4.2–8.3)	3.0 (1.8–4.9)	−0.59 (−1.09–−0.09)	12.5 (7.6–19.8)	4.7 (1.7–12.3)	−1.57 (−3.09–−0.05)	4.3 (1.9–9.6)	1.6 (0.4–6.1)	−0.55 (−1.37–0.28)	−1.02 (−2.75–0.71)
Dominican Republic	DHS 2002	MICS 2014	5.0 (4.0–6.2)	8.9 (7.7–10.2)	0.32 (0.18–0.46)	9.4 (7.2–12.2)	14.1 (11.5–17.1)	0.39 (0.07–0.70)	4.8 (2.3–9.6)	6.3 (4.1–9.7)	0.13 (−0.24–0.49)	0.26 (−0.22–0.74)
Gabon	DHS 2000	DHS 2012	30.6 (27.0–34.5)	11.6 (9.0–14.7)	−1.59 (−1.98–−1.20)	50.8 (45.2–56.4)	18.1 (13.9–23.3)	−2.72 (−3.33–−2.12)	21.2 (13.7–31.3)	8.5 (2.3–26.5)	−1.06 (−2.19–0.08)	−1.67 (−2.95–−0.38)
Guyana	MICS 2006	MICS 2014	6.3 (4.0–10.0)	4.4 (2.7–7.2)	−0.23 (−0.69–0.22)	8.2 (4.1–15.7)	6.3 (3.6–10.7)	−0.24 (−1.05–0.57)	2.1 (0.3–13.7)	1.9 (0.5–7.3)	−0.03 (−0.63–0.57)	−0.21 (−1.22–0.80)
Jordan	DHS 2007	DHS 2017	1.1 (0.6–2.3)	7.4 (5.7–9.7)	0.63 (0.41–0.84)	1.3 (0.4–3.7)	9.4 (6.3–13.9)	0.82 (0.42–1.21)	2.0 (0.3–12.4)	7.5 (2.5–20.0)	0.55 (−0.32–1.41)	0.27 (−0.68–1.22)
Kazakhstan	MICS 2006	MICS 2015	0.9 (0.4–2.0)	4.4 (3.2–6.0)	0.38 (0.21–0.56)	0.9 (0.2–3.6)	3.6 (1.6–7.6)	0.29 (−0.04–0.63)	1.0 (0.1–6.9)	8.0 (3.9–15.7)	0.77 (0.12–1.43)	−0.48 (−1.22–0.26)
Maldives	DHS 2009	DHS 2016	1.2 (0.6–2.4)	9.2 (6.6–12.8)	1.15 (0.70–1.60)	1.0 (0.2–4.3)	9.5 (6.0–14.8)	1.21 (0.56–1.87)	1.3 (0.2–8.2)	16.4 (6.7–34.8) *	2.16 (0.18–4.14)	−0.95 (−3.03–1.13)
Namibia	DHS 2006	DHS 2013	5.3 (3.8–7.4)	7.3 (5.5–9.7)	0.29 (−0.11–0.68)	10.2 (5.8–17.2)	2.6 (1.2–5.6)	−1.08 (−1.92–−0.25)	1.8 (0.6–4.8)	12.7 (7.4–21.0)	1.56 (0.57–2.55)	−2.65 (−3.94–−1.35)
North Macedonia	MICS 2005	MICS 2018	10.6 (6.9–16.0)	4.1 (1.7–9.5)	−0.50 (−0.94–−0.07)	12.6 (8.9–17.5)	0.0 (0.0–0.0)	−0.97 (−1.30–−0.64)	18.2 (10.0–30.9)	0.6 (0.1–4.3)	−1.35 (−2.15–−0.55)	0.38 (−0.48–1.25)
Peru	DHS 2009	DHS 2019	4.3 (3.2–5.6)	4.7 (3.9–5.7)	0.05 (−0.10–0.20)	4.3 (2.8–6.6)	6.3 (4.7–8.3)	0.19 (−0.07–0.45)	2.4 (0.5–10.5)	5.0 (2.6–9.2)	0.26 (−0.23–0.74)	−0.06 (−0.61–0.48)
Serbia	MICS 2005	MICS 2019	4.3 (2.9–6.5)	3.5 (1.8–6.7)	−0.06 (−0.26–0.15)	8.7 (4.7–15.6)	7.6 (2.9–18.6)	−0.08 (−0.71–0.56)	3.4 (1.1–9.9)	0.5 (0.1–3.9)	−0.20 (−0.48–0.07)	0.13 (−0.56–0.82)
Suriname	MICS 2006	MICS 2018	10.1 (7.3–13.9)	19.7 (15.5–24.7)	0.80 (0.33–1.27)	9.0 (4.9–15.9)	22.7 (16.3–30.8)	1.14 (0.39–1.89)	12.4 (6.4–22.4)	21.6 (10.8–38.8)	0.77 (−0.56–2.11)	0.37 (−1.16–1.89)
Thailand	MICS 2005	MICS 2019	2.5 (1.7–3.8)	3.1 (1.8–5.4)	0.04 (−0.10–0.18)	3.4 (1.9–6.2)	3.1 (1.8–5.4)	−0.03 (−0.22–0.17)	2.4 (0.6–9.4)	0.1 (0.1–0.4)	−0.16 (−0.41–0.08)	0.14 (−0.17–0.45)
Turkey	DHS 2003	DHS 2013	11.5 (9.0–14.7)	5.6 (3.7–8.3)	−0.59 (−0.96–−0.23)	31.0 (23.6–39.6)	7.2 (3.9–12.8)	−2.38 (−3.28–−1.48)	1.1 (0.2–5.2)	3.3 (0.9–11.5)	0.22 (−0.23–0.68)	−2.60 (−3.61–−1.59)

Note: The absolute annual change was calculated by subtracting prevalence in year 1 from prevalence in year 2 and dividing by the number of intervening years. The absolute annual excess change was calculated by subtracting the annual absolute change in quintile 5 (richest) from the annual absolute change in quintile 1 (poorest). Countries classified according to 2021 World Bank country income level groupings. DHS = Demographic and Health Survey. DTP = diphtheria–tetanus–pertussis vaccine. CI = confidence interval. MICS = Multiple Indicator Cluster Survey. * Estimate is based on 25–49 cases.

## Data Availability

The data in this study are contained within the article.

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
