# Peer review of "Economic-Related Inequalities in Zero-Dose Children: A Study of Non-Receipt of Diphtheria–Tetanus–Pertussis Immunization Using Household Health Survey Data from 89 Low- and Middle-Income Countries"

_vaccines, 2022, doi:10.3390/vaccines10040633_

Round 1

Reviewer 1 Report

This is a really interesting and important paper for the Vaccination community. The article reads well, and the analysis done for a significant number of markets makes this manuscript a very interesting one. Specially because this paper highlights a really concerning problem as the children with zero doses of DTP around the world with a special focus on the low-income countries. I truly found it of interest and not much to add on the methodology and results interpretation.

I'd only request the authors to provide in the discussion section, more information on which markets assessed are on track to achieve the Gavi 5.0 goals by 2025 (25% reduction) and potentially to 2030. The reason is that some markets had made progress (we all know that others don't) but that progress could still be insufficient towards the milestones proposed for those markets. In case, none of these markets is expected to meet the reduction target this will be an interesting finding as well to report. In addition, I feel that authors may include the definitions for the index of inequality and the concentration index, these are not found in the manuscript text at present. The latter should provide more transparency to the article and its findings.

Author Response

Thank you for your feedback on our submission.

Regarding your request for “more information on which markets assessed are on track to achieve the Gavi 5.0 goals”:

Both Gavi 5.0 and IA2030 rely on the WHO/UNICEF Estimates of National Immunisation Coverage (WUENIC) to measure progress towards 2025 and 2030 targets. As these strategies began in 2021, and the latest WUENIC estimates (published July 2021) only provide estimates of vaccination coverage and zero-dose children through 2020, it is too early to comment on which countries or regions are on track towards the targets. The most recent WUENIC estimates did document disruptions to routine immunisation coverage in 2020 due to COVID-related disruptions (which we acknowledge in the introduction of our paper). Whether those disruptions continued into 2021 or later is a key question, and we will have more insight when WUENIC estimates through 2021 are published in July of this year. As the survey data in this analysis are historical, and vaccination coverage can change quickly at country level, we do not think it is appropriate to comment on progress towards 2021-2025 and 2030 targets in the paper. We have added the following sentence to the first paragraph of the discussion section, highlighting the importance of the equity agenda of Gavi 5.0 and IA2030: “The stark increase in the number of zero-dose children in 2020 further underscores the importance of the equity orientation of Gavi 5.0 and IA2030 and the need to restore, strengthen and expand the reach of immunization systems.”

Reviewer 2 Report

In this study, the authors used 2000-2019 household survey data from 154 surveys representing 89 low- and middle-income countries to assess within-country, economic-related inequality in the prevalence of one-year-old children with zero doses of diphtheria-tetanus-pertussis (DTP) vaccine. The difference, ratio, slope index of inequality, concentration index, and excess change measures were calculated to assess the latest situation and change over time, by country-income grouping and for 17 countries with high zero-dose DTP numbers and prevalence.

Across 89 countries, the median prevalence of zero-dose DTP was 7.6%. Low-income countries had higher inequality than lower-middle-income countries and upper-middle-income countries.

They concluded that zero-dose DTP prevalence among the poorest households of low-income countries declined between 2000-2009 and 2010-2019, yet economic-related inequality remained high in many countries. Economic-related inequalities in zero-dose DTP prevalence are particularly pronounced in low-income countries.

The study is of interest and provides novel findings.

I have only comment regarding Materials and methods: please specify for each country considered the local DTP vaccine policy, in particular, whether it is only recommended or mandatory. This should be considered in analyzing differences.

Author Response

Thank you for your feedback on our submission. Regarding your request to “specify for each country considered the local DTP vaccine policy…”:

This information is not readily available. Since the inception of the Expanded Programme on Immunization in the 1970s, WHO recommends that all countries have at least three DTP doses in their schedule (which all countries have). We have added this point to the discussion section of the paper (see line 336). In our discussion we further outline reasons why children from disadvantaged subgroups may more often lack access to vaccinations, citing previous research findings about women’s social status, access barriers and health system factors, among other determinants of health.

Reviewer 3 Report

A nicely written paper. One minor issue

Line 127 - a stray bracket at the end

Are you able to do some analysis of country average versus factors such as proportion of the population living in rural areas, proportion of the population living in slums, and GDP per head. All data readily available. Perhaps in an appendix.

Perhaps also show a chart of the situation in the richest quintile across all countries and some comment.

Author Response

Thank you for your feedback on our submission. Please note the following responses to the comments provided:

  • Typo on line 127 has been corrected.

Regarding the suggestion to do an analysis of country average versus other factors:

  • The proposed analysis could be done to assess the associations between the suggested factors and national average of zero-dose DTP, though this is outside the scope of the current paper, which focuses on quantifying and comparing the level of inequality across countries.

Regarding the request for a chart of the situation in the richest quintile:

  • This information is available in Figure 1 and Table 1. We have added a comment about this pattern (line 178): “Across all countries, the range of prevalence values in the richest quintile was smaller than the range of values in the poorer quintiles.”